# Unexpected Tension Pneumothorax Developed during Anesthetic Induction Aggravated by Positive Pressure Ventilation: A Case Report

**DOI:** 10.3390/medicina59091631

**Published:** 2023-09-08

**Authors:** Seunghee Ki, Beomseok Choi, Seung Bae Cho, Seokwoo Hwang, Jeonghan Lee

**Affiliations:** Department of Anesthesiology and Pain Medicine, Busan Paik Hospital, Inje University College of Medicine, Busan 47392, Republic of Korea; 108873@paik.ac.kr (S.K.); 089535@paik.ac.kr (B.C.); 089517@paik.ac.kr (S.B.C.); 094157@paik.ac.kr (S.H.)

**Keywords:** central venous catheterization, intraoperative monitoring, mechanical ventilation, positive-pressure ventilation, tension pneumothorax

## Abstract

*Background and Objectives:* Tension pneumothorax is a life-threatening emergency condition that requires immediate diagnosis and intervention. However, due to the non-specific symptoms and the rarity of its occurrence during surgery, anesthesiologists encounter difficulties in promptly diagnosing tension pneumothorax when it arises intraoperatively. Diagnosing tension pneumothorax can become even more challenging in unexpected situations in patients with normal preoperative evaluation for general anesthesia. *Materials and Methods*, *Results:* We report the case of a 66-year-old woman who underwent general anesthesia for oblique lateral interbody fusion surgery of her lumbar spine. Though she did not have any respiratory symptoms prior to the induction of anesthesia, auscultation following endotracheal intubation indicated decreased breathing sound in the left hemithorax of the chest. Subsequently, her vital signs showed tachycardia, hypotension, and hypoxemia, and the ventilator indicated a gradual increase in the airway pressure. We verified the proper depth of the endotracheal tube to exclude one-lung ventilation, and, in the meantime, learned that there had been unsuccessful attempts at left subclavian venous catheterization by the surgical department on the previous day. Tension pneumothorax was diagnosed through portable chest radiography in the operating room, and needle thoracostomy and chest tube insertion were performed immediately, which in turn stabilized her vital signs and airway pressure. The surgery was uneventful, and the chest tube was removed one week later after evaluation by the cardiothoracic department. The patient was discharged from hospital on postoperative day 14 without known complications. *Conclusions:* Anesthesiologists should be aware of the conditions and risk factors that may cause tension pneumothorax and remain vigilant for signs of its development throughout surgery, even for patients who show normal preoperative assessments. An undetected small pneumothorax without any symptoms can progress to tension pneumothorax through positive pressure ventilation during general anesthesia, posing a life-threatening situation. If a tension pneumothorax is highly suspected through clinical assessments, its prompt differentiation and timely diagnosis are crucial, allowing for rapid intervention to stabilize vital signs.

## 1. Introduction

A pneumothorax is a medical condition consisting in the presence of air between the parietal and visceral pleural cavity. Even a small simple pneumothorax can worsen during general anesthesia due to positive pressure mechanical ventilation rapidly increasing its size and severity, which can lead to tension pneumothorax [1]. Tension pneumothorax is a life-threatening emergency condition that requires immediate diagnosis and intervention. However, since tension pneumothorax rarely occurs during general anesthesia, it may be unfamiliar to the anesthesiologist in the operating room [2]. In particular, in cases where tension pneumothorax occurs unexpectedly in patients who manifest non-specific clinical symptoms, diagnosis may be delayed [1]. Consequently, delayed diagnosis and intervention can lead to adverse and potentially devastating consequences [3].

We present a case of unexpected tension pneumothorax in a patient who had no underlying respiratory disease or related symptoms prior to the induction of general anesthesia. The development of pneumothorax was later discovered to be iatrogenic from multiple subclavian venous catheterization attempts by the surgical department one day prior to the scheduled surgery, which succeeded our pre-anesthetic evaluations. For this reason, diagnosing tension pneumothorax was challenging after tracheal intubation when the patient’s vital signs gradually worsened because we were not aware of those failed attempts. This case report presents a detailed description of the diagnostic process of tension pneumothorax, while emphasizing the importance of pre-anesthetic evaluation for all events prior to the patient entering the operating room.

## 2. Case Description

A 66-year-old-woman (149 cm in height, 51 kg in weight), who had been suffering from worsening back pain and sciatica of the left leg for two years, presented herself at the neurosurgery spine center of our hospital and was diagnosed with spinal stenosis on her L3/4/5. She was scheduled to undergo oblique lateral interbody fusion (OLIF) surgery under general anesthesia. Her medical history included hypertension and rheumatoid arthritis, which was under control through medication. Preoperative evaluations were performed, and laboratory values were all within normal range. No abnormalities were observed in her lungs in the chest radiography taken one week before the scheduled surgery (Figure 1). She was hospitalized one day before surgery; however, securing her peripheral venous access was challenging. Considering the possibility of prolonged surgery requiring significant fluid and medication administration, the neurosurgery department proceeded with central venous catheter (CVC) insertion in the general ward prior to surgery. The neurosurgeon attempted an infraclavicular approach to the left subclavian vein under ultrasound guidance; however, two of the attempts failed. Then, the CVC was successfully inserted in the right subclavian vein after one attempt. After the procedure, no chest radiograph was taken to rule out any complications or to verify the proper placement of the CVC. Gauge dressing was applied above the needle insertion points beneath her left clavicle for hemostasis; however, it was removed before she was transferred to the operating room on the day of the surgery.

The patient was premedicated intramuscularly with glycopyrrolate (0.2 mg) and famotidine (20 mg) 30 min prior to entering the operating room. When she entered the operating room, her initial vital signs were monitored: non-invasive blood pressure (NIBP) 161/108 mmHg, heart rate 126 beats/min, and SpO_2_ 95%. Her electrocardiography did not show any particular abnormality other than tachycardia. We questioned her tachycardia and the lower-than-expected SpO_2_, and asked the patient whether she had uncomfortable symptoms. She complained of preoperative anxiety and thirst but denied any other symptoms such as dyspnea and chest pain. Her surgery was scheduled for the afternoon, and she had fasted since midnight. We suspected that her tachycardia was caused by a combination of dehydration due to NPO time of more than 12 h, preoperative anxiety, and the effect of premedicated glycopyrrolate. We had difficulty anticipating the cause of the low SpO_2_, but as she did not complain of dyspnea, we considered it insignificant. While conducting preoxygenation at fraction of inspired oxygen (FiO_2_) of 1.0, we administered crystalloid fluids to correct her dehydration. During the 5 min preoxygenation period, the SpO_2_ increased to 100%, approximately 100 cc of crystalloid fluid was infused, and her heart rate decreased to 98 beats/min. General anesthesia was induced with propofol (100 mg) and infusion of remifentanil (0.5 µg/kg/min). To facilitate the endotracheal tube insertion, rocuronium bolus (30 mg) was injected. We performed manual mask ventilation with a tidal volume of 300 to 500 mL before tracheal intubation. The peak inspiratory pressure (PIP) remained below 25 mmHg throughout manual ventilation. After an electromyograph confirmed deep neuromuscular block, tracheal intubation using 7.0 Fr endotracheal tube (Covidien Ireland Limited, Tullamore, Ireland) was successfully performed without much difficulty and the grade of modified Cormack-Lehane scale was 1. After intubation, we performed auscultation on each hemithorax to confirm the success of endotracheal intubation and to exclude the possibility of endobronchial intubation. In the process, we discovered that the breathing sound in the left hemithorax was excessively decreased, and we adjusted the depth of the intubated tube as we suspected right endobronchial intubation. However, the left hemithorax breathing sound remained significantly decreased. Therefore, we decided to check for the proper depth of the endotracheal tube using a flexible fiberoptic bronchoscope while temporarily fixing the tube at a depth of 20 cm from the upper incisors. While preparing to use the flexible fiberoptic bronchoscope, mechanical ventilation was initiated with a FiO_2_ value of 0.5, a tidal volume of 350 mL, a respiratory rate of 12/min, and positive end expiratory pressure (PEEP) of 5 mmHg on the volume control AutoFlow (VC-AF) mode of the ventilator (Primus^®^, Dräger, Lübeck, Germany). The end tidal CO_2_ was 31 mmHg, and the PIP was 18 mmHg. Anesthesia was maintained with inhalation of sevoflurane (2.0%) and infusion of remifentanil (0.25 µg/kg/min). Placing the flexible fibrotic bronchoscope (iS3-F^®^, Insighters, Shenzhen, China) in the endotracheal tube, we visually confirmed that the distal end of the endotracheal tube was approximately 3 cm above the carina, confirming that the tube was in an appropriate position and the decreased breathing sound of her left hemithorax was not due to endobronchial intubation. No bronchial constriction or foreign body that could cause the decreased breathing sound of the left hemithorax was observed. During evaluation for other causes of the observed clinical signs, a progressive deterioration in the patient’s vital signs was noted, manifesting as a gradual decline in SpO_2_ to 94%, a reduction in NIBP to 74/42 mmHg, and an elevated heart rate of 116 beats/min. The PIP increased to 25 mmHg. To stabilize her vital signs, we raised FiO_2_ from 0.5 to 0.8, administered phenylephrine bolus (50 µg), and started phenylephrine infusion (0.5 µg/kg/min). After her blood pressure had stabilized, we conducted a physical examination again to determine the cause of this situation and discovered two pinhole-sized scabs beneath her left clavicle. Medical records confirmed that they were caused by attempts at left subclavian venous catheterization on the previous day. We strongly suspected that the cause of these clinical signs correlated with iatrogenic pneumothorax, and a portable chest radiograph was immediately taken. Chest radiography revealed the presence of a tension pneumothorax characterized by a notable mediastinal shift towards the right, diaphragmatic depression, and profound collapse of the left lung (Figure 2A). We promptly adjusted the ventilator setting (FiO_2_ of 0.8, a tidal volume of 300 mL, a respiratory rate of 14/min, no PEEP, and VC-AF mode) to prevent further aggravation of the tension pneumothorax. Emergency needle thoracostomy was performed, followed by the insertion of a chest tube. Portable chest radiography after the insertion of a chest tube revealed that the pneumothorax had significantly resolved (Figure 2B). After the chest tube insertion, PIP decreased to 20 mmHg, NIBP increased to 142/80 mmHg, and heart rate decreased to 100 beats/min. Subsequently, even upon discontinuation of phenylephrine infusion and lowering FiO_2_ to 0.5, the patient showed stable vital signs, all within the normal range. Since her vital signs were stable and the resolution of the pneumothorax was confirmed by chest radiography, we determined to proceed with the surgery while maintaining the chest tube in place. 

The surgery lasted 7 h and was completed uneventfully. We successfully performed extubation and postoperatively the patient did not complain of any symptoms other than localized pain at the surgical site. She was transferred to the general ward after recovery from anesthesia. Chest radiography on postoperative day 7 revealed that the pneumothorax had almost resolved (Figure 3). Following confirmation from the cardiothoracic department, the chest tube was removed on the same day. On postoperative day 14, she was discharged from hospital without any observed complications.

## 3. Discussion

Pneumothorax during anesthesia and surgery has been previously reported, but it still remains a relatively rare perioperative event [2]. A more recent study confirms that pneumothorax is a less frequent event when compared to other complications associated with anesthesia [4]. Factors associated with the development of pneumothorax include trauma and patient’s underlying pulmonary diseases, such as asthma, COPD, and emphysematous bullae which are vulnerable to rupture due to positive pressure ventilation [5,6,7]. Additionally, iatrogenic factors, including surgical and anesthetic procedures, such as pneumoperitoneum for laparoscopic procedures, CVC insertion, regional blocks adjacent to the pleura, and incorrect endotracheal intubation can also contribute to its occurrence [2,5,8]. In our case, there were no factors relating to pneumothorax other than the CVC insertion and positive pressure ventilation. Since the patient had no underlying pulmonary disease, we strongly presume that the pneumothorax had developed from the failed attempts on CVC insertion in the left subclavian vein. However, due to the absence of chest CT imaging, we cannot completely exclude the possibility of pneumothorax caused by ruptured bullae. We speculate that positive pressure ventilation during the process of anesthesia induction and mechanical ventilation may have exacerbated its progression, leading to tension pneumothorax. Positive pressure ventilation is a well-known factor associated with pneumothorax, and it can exacerbate even a small air leak or asymptomatic pneumothorax, potentially progressing into tension pneumothorax [1,6,9,10,11,12].

The subclavian CVC, when compared to the internal jugular and femoral CVC, has a lower risk of infection and thrombosis, but a higher risk of iatrogenic pneumothorax [13]. The risk of iatrogenic pneumothorax increases in instances when CVC insertion requires three or more needle passes, encounters procedural failure at the initial vein site, or involves positive pressure ventilation [10,14]. Considering these risk factors, the patient in our case was at high risk of iatrogenic pneumothorax. Unfortunately, chest radiography after CVC insertion was not performed to rule out any complications or to confirm the proper placement of the CVC tip. We also learned about the failed CVC insertion attempts after the patient was under general anesthesia when her vital signs became unstable during mechanical ventilation. The neurosurgeon stated that the patient’s pleura was intact without needle penetration during CVC insertion attempts. Furthermore, the patient did not exhibit any symptoms, such as chest pain, shortness of breath or respiratory distress, after the procedure; therefore, he omitted an immediate chest radiograph, but instead planned a chest radiograph postoperatively for lung evaluation and the correct placement of the CVC tip. According to our hospital protocol, acquiring a chest radiograph following CVC insertion is recommended but not mandatory. However, due to the broad spectrum of clinical presentations for pneumothorax, the lack of symptoms cannot definitively exclude the possibility of pneumothorax in patients. In addition, the CVC insertion was performed on the evening prior to the surgery, which succeeded the patient’s pre-anesthesia evaluation that had been conducted in the afternoon. Therefore, we were unaware of the failure of the patient’s left subclavian catheterization before the induction of general anesthesia. These two circumstances led to an incomplete pre-anesthesia evaluation of pneumothorax in a patient who should have been regarded as being at high risk of pneumothorax.

Diagnosing pneumothorax can be challenging because it presents with diverse and non-specific clinical symptoms, necessitating the exclusion of other potential causes that can produce similar manifestations [1,15]. The patient in our case had tachycardia and lower-than-expected SpO_2_ when she entered the operating room, which could be considered signs of pneumothorax. However, it is not possible to suspect pneumothorax based on these signs alone. Various factors can influence the patient’s heart rate and SpO_2_, and heart rate and pulse oximetry are considered non-specific signs in pneumothorax [1]. After performing endotracheal intubation on the patient, we noticed a decreased breathing sound on the left hemithorax, which could be considered a typical indicator of pneumothorax. However, a decreased breathing sound in an anesthetized patient, particularly on the left side, is typically due to mainstem bronchus intubation [1]. Thus, before considering pneumothorax as a possibility, we decided to confirm the presence of endobronchial intubation. We employed a flexible fiberoptic scope, but a portable X-ray had already been prepared in the operating room for the OLIF surgery. If we had considered performing a portable chest radiography to confirm the position of the tube, we might have detected the pneumothorax earlier.

Taking a chest radiography in patients under general anesthesia differs from of the procedure in conscious patients. Because it is challenging to take an erect posteroanterior (PA) chest radiograph in the supine position for anesthetized patients, a supine anteroposterior (AP) chest radiograph is more frequently taken in the operating room. It is important to note that the supine AP chest radiograph may exhibit dissimilar images to those observed on a familiar erect PA chest radiograph, and some findings on a normal chest radiograph can be confused with pneumothorax [6,15]. These factors can pose a challenge for anesthesiologists and physicians in diagnosing pneumothorax, even with the use of chest radiographs. Despite its limitations, we decided to use portable radiography because it was already in place and ready in the operating room for the OLIF surgery. 

Transthoracic sonography is another tool that is helpful not only in diagnosing pneumothorax, but also in excluding other life-threatening conditions. Compared to supine chest radiography, transthoracic sonography offers several advantages, including lower radiation exposure, portability, real-time imaging capabilities, more sensitivity, and the ability to perform dynamic and repeat evaluation [6,16,17]. In most bedside circumstances, when considering the time required for the preparation and execution of a portable chest radiograph, a prompt chest evaluation by transthoracic sonography may offer superior benefits in the diagnosis of pneumothorax [17]. It can also be utilized immediately for the treatment of pneumothorax to guide a needle thoracostomy or chest tube insertion.

One aspect we should contemplate is whether we could have detected the pneumothorax before the patient entered the operating room if we had obtained a preoperative chest radiograph. A routine chest radiography after CVC insertion is recommended to rule out complications such as pneumothorax and catheter misplacement [18]. However, immediate chest radiography following CVC insertion often exhibits limitations in accurately diagnosing pneumothorax, potentially resulting in false safety and diverting attention from important signs of complications [9,11,12,14,19]. Considering the economic costs, radiation exposure, and the relatively low incidence of pneumothorax as a complication following CVC insertion, the necessity of routine chest radiography after CVC insertion can be controversial [14,18,19]. Nevertheless, clinicians who argue that routine chest radiography after CVC insertion is not needed still recommend screening tests such as chest radiography for individuals at high risk of complications [14,18,19]. Given these considerations, although we cannot guarantee that post-procedural chest radiography would have revealed the pneumothorax, our patient necessitated it.

## 4. Conclusions

Even a small and asymptomatic pneumothorax can progress to tension pneumothorax through positive pressure ventilation, posing a life-threatening situation. Anesthesiologists should be aware of the risk factors related to pneumothorax in patients before induction of general anesthesia. Patients at high risk of pneumothorax require thorough preoperative evaluation, and even if the preoperative assessment appears normal, anesthesiologists should be able to anticipate the occurrence of pneumothorax during the surgery. When clinical symptoms suggestive of a tension pneumothorax arise, its prompt differentiation from other causes and timely diagnosis are essential, allowing for rapid intervention to stabilize vital signs.

## Figures and Tables

**Figure 1 medicina-59-01631-f001:**
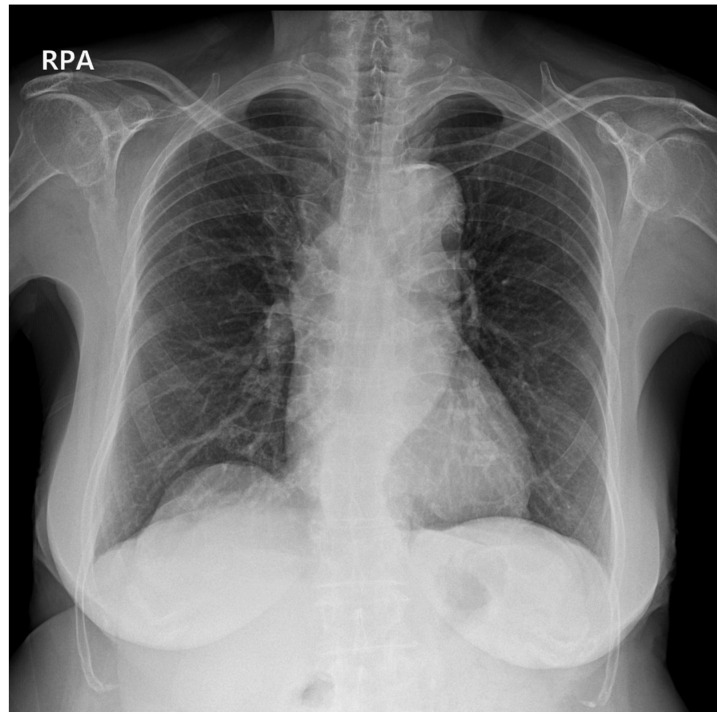
Posteroanterior (PA) chest radiograph taken as a routine preoperative evaluation was without any abnormal findings in the lungs, mediastinum, or thoracic cavity.

**Figure 2 medicina-59-01631-f002:**
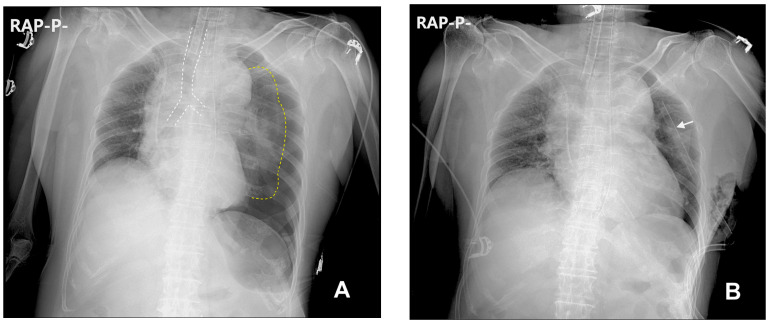
Supine anteroposterior (AP) chest radiograph obtained in the operating room with portable radiography shortly after the patient’s vital signs had become unstable: (**A**) a tension pneumothorax of the left lung is prominent with mediastinal shift towards the right (trachea in white dotted lines), left diaphragmatic depression, and profound collapse of the left lung (yellow dotted lines); (**B**) the pneumothorax resolved shortly after needle thoracotomy and chest tube (white arrow) insertion.

**Figure 3 medicina-59-01631-f003:**
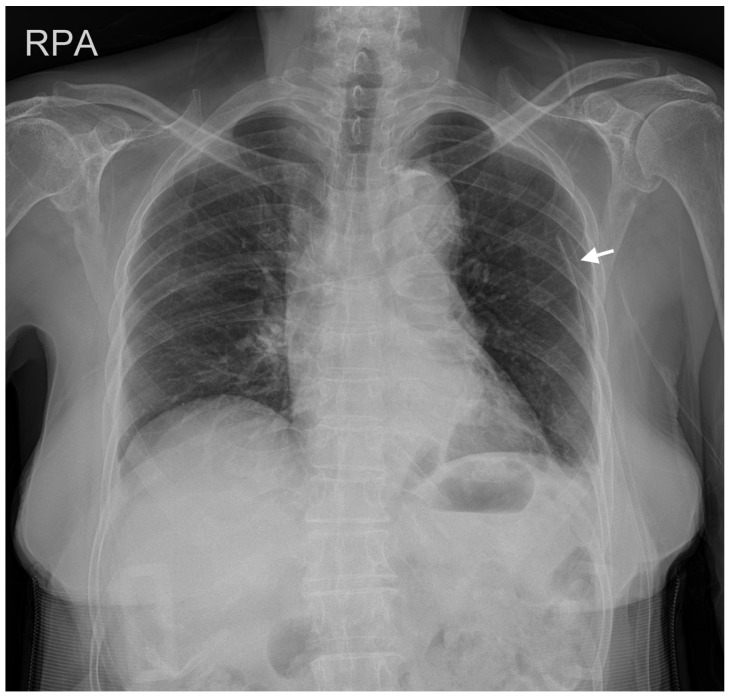
PA chest radiograph on postoperative day 7 with chest tube (white arrow) in place. The pneumothorax is no longer visible, and the mediastinum is similar to that of the preoperative PA chest radiograph (Figure 1).

## Data Availability

Not applicable.

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
