# Peer review of "Unexpected Tension Pneumothorax Developed during Anesthetic Induction Aggravated by Positive Pressure Ventilation: A Case Report"

_medicina, 2023, doi:10.3390/medicina59091631_

Round 1
Reviewer 1 Report
It's an interesting case of tension pneumothorax. Although it is not a rare case of tension pneumothorax, I found it interesting due to its obscure etiology. Tension pneumothorax is a challenging clinical condition, especially in patients without apparent risk factors. Bedside ultrasound examination is helpful not only to diagnose it but to exclude other life-threatening conditions with similar clinical signs, so all clinical specialties should be familial with the detection of tension pneumothorax using US. The study is well written. The discussion paragraph could be less detailed and focused only on significant points.
Author Response
Thank you for your time in reviewing our manuscript.
Minor changes have been made to the manuscript by removing less significant details in our discussion. We have also rearranged part of the discussion regarding chest radiography and sonography to make it more focused on significant points.
Please refer to the revised manuscript with changes (highlighted yellow). Thank you.
Reviewer 2 Report
This manuscript describes a clinical case of tension pneumothorax after induction of anaesthesia. This complication is not "exotic" and should be considered in any cause of haemodynamic instability with ventilation problems at an early stage.
The case report is detailed, complete and well presented, the discussion contains all essential aspects.
From my point of view, there are no suggestions for improvement of this manuscript.
Author Response
Thank you for your time in reviewing our manuscript.
According to another reviewer’s opinion, minor revision has been made to the manuscript (highlighted in yellow) to focus on more significant points in the discussion section.